# Extrahepatic Cancer Risk in Patients with Hepatitis C Virus Infection Treated with Direct-Acting Antivirals

**DOI:** 10.3390/microorganisms12091926

**Published:** 2024-09-22

**Authors:** Joji Tani, Tsutomu Masaki, Kyoko Oura, Tomoko Tadokoro, Asahiro Morishita, Hideki Kobara

**Affiliations:** 1Department of Gastroenterology and Neurology, Faculty of Medicine, Kagawa University, 1750-1 Ikenobe, Miki, Kita, Takamatsu 761-0793, Kagawa, Japan; oura.kyoko@kagawa-u.ac.jp (K.O.); tadokoro.tomoko@kagawa-u.ac.jp (T.T.); morishita.asahiro@kagawa-u.ac.jp (A.M.); kobara.hideki@kagawa-u.ac.jp (H.K.); 2Kagawa Saiseikai Hospital, Takamatsu 761-8076, Kagawa, Japan; tmasaki@saiseikai-kagawa.jp

**Keywords:** hepatitis C virus, direct-acting antivirals, extrahepatic cancer, sustained virologic response

## Abstract

Chronic hepatitis C virus (HCV) infection is associated with an increased risk of extrahepatic cancers, particularly non-Hodgkin lymphoma. The introduction of direct-acting antivirals (DAAs) has revolutionized HCV therapy, resulting in high cure rates. However, concerns have been raised about potential effects on cancer risk. This review summarizes the current evidence on extrahepatic cancer risk in HCV-infected patients treated with DAAs. We examined epidemiologic data on HCV-associated extrahepatic cancers and explored potential mechanisms linking HCV to carcinogenesis outside the liver. Studies evaluating cancer outcomes after DAA therapy were critically reviewed while considering methodological challenges. While some studies suggested a reduced risk of extrahepatic cancers after DAA therapy, others showed no significant change. Limitations included short follow-up periods and confounding variables. Immunological changes following rapid HCV clearance may have complex effects on cancer risk. Long-term prospective studies and mechanistic investigations are needed to fully elucidate the relationship between DAA therapy and extrahepatic cancer risk in HCV patients. Clinicians should remain vigilant for extrahepatic malignancies in this population.

## 1. Introduction

Chronic hepatitis C virus (HCV) infection represents an important global health challenge, affecting approximately 71 million individuals worldwide and causing substantial morbidity and mortality [1]. While HCV is primarily recognized for its deleterious effects on the liver, there is mounting evidence of its far-reaching impact beyond hepatic tissues. Extrahepatic manifestations, which occur in an estimated 40–70% of patients with chronic HCV infection, encompass a wide spectrum of conditions ranging from mild, subclinical abnormalities to severe, life-threatening systemic diseases [2,3]. These diverse manifestations can include immune-related disorders, lymphoproliferative diseases, metabolic alterations, and various types of extrahepatic cancers, collectively contributing to the overall disease burden associated with HCV infection [4,5]. Moreover, a comprehensive analysis highlighted the important contribution of HCV infection to the global burden of infection-attributable cancers, emphasizing the need for a thorough understanding of the role of HCV infection in both hepatic and extrahepatic malignancies [6].

HCV therapy underwent a revolutionary transformation in 2014 with the introduction of direct-acting antiviral agents (DAAs). These innovative therapies offer remarkably high cure rates exceeding 95%, coupled with short therapy durations and minimal side effects, and represent a paradigm shift from the previous interferon-based regimens [7]. This therapeutic breakthrough has not only dramatically improved liver-related outcomes, but has also shown promise in mitigating various extrahepatic manifestations of HCV infection. The potential reduction in extrahepatic cancer risk is of particular interest, as this could significantly impact the long-term prognosis and quality of life for individuals with a history of HCV infection [8,9]. However, the rapid and profound virological suppression achieved by DAAs has raised questions about potential unintended consequences. Some researchers have expressed concerns about the immunologic changes caused by DAAs and the possibility of decreased tumor control in the immediate post-treatment period [10,11]. These concerns arise from early observations of unexpected occurrences or recurrences of hepatocellular carcinoma (HCC) in some patients shortly after DAA therapy. While subsequent larger studies have largely allayed these concerns for liver cancer, the potential impact on extrahepatic cancer risk remains an area of active investigation and debate within the scientific community [12,13,14,15,16].

Given the complexities of HCV pathogenesis and its systemic effects, a comprehensive understanding of the relationship between DAA therapy and extrahepatic cancer risk is crucial to optimize patient care and public health strategies. This review aimed to synthesize the current evidence regarding the impact of DAA therapy on extrahepatic cancer risk in HCV-infected patients, and provide a critical analysis of the available data and highlight areas that require further research.

This review begins by examining the epidemiological data on HCV-associated extrahepatic cancers, exploring the prevalence and risk factors for various malignancies in the HCV-infected population. This is followed by an in-depth exploration of the potential mechanisms linking HCV infection to carcinogenesis outside the liver, including chronic immune stimulation, oxidative stress, metabolic alterations, and direct oncogenic effects of viral proteins. We then critically evaluated studies assessing cancer outcomes following DAA therapy, comparing and contrasting findings from large-cohort studies, registry-based investigations, and meta-analyses. We discussed the challenges inherent in studying extrahepatic cancer risk in this population, including issues related to confounding factors, competing risks, and the need for long-term follow-up. Additionally, we explored the immunological considerations surrounding DAA therapy and its potential implications for cancer risk. This included an examination of the changes in immune function, cytokine profiles, and tumor surveillance mechanisms that occur during and after HCV eradication with DAAs. Finally, we highlighted gaps in the current knowledge and outlined priority areas for future research. This encompassed recommendations for long-term follow-up studies, cancer-specific investigations, and mechanistic studies to better understand the complex interplay between HCV infection, DAA therapy, and extrahepatic cancer risk.

By comprehensively addressing these aspects, this review aimed to provide clinicians, researchers, and policymakers with a nuanced understanding of the current state of knowledge regarding extrahepatic cancer risk in HCV-infected patients treated with DAAs. This information is critical to developing evidence-based guidelines for cancer surveillance and prevention in the growing population of individuals with a history of HCV infection who have achieved sustained virologic response (SVR) through DAA therapy.

## 2. Materials

For this review, the authors performed a comprehensive search for relevant studies from available peer-reviewed journals using electronic databases, including PubMed and MDPI. After the initial database search, the list of identified articles was reviewed to select articles that potentially met the eligibility criteria. The full text for selected articles that met the eligibility criteria was then accessed. Articles that were not in English or were deemed inappropriate were excluded; a final 107 articles were selected.

## 3. Epidemiology of HCV-Associated Extrahepatic Cancers (Table 1)

HCV infection has been consistently associated with an increased risk of several extrahepatic cancers, most notably non-Hodgkin lymphoma (NHL) and intrahepatic cholangiocarcinoma [17,18]. Recent large-scale epidemiological studies have further strengthened the evidence of associations between HCV infection and various cancer types. A registry-based case–control study of older adults in the USA revealed an increased risk for several types of cancer in HCV-infected individuals [19,20].

### 3.1. Hematologic Malignancies

NHL is the most extensively studied extrahepatic malignancy associated with HCV infection. A seminal meta-analysis by Dal Maso and Franceschi found that HCV-infected individuals had a 2–3-fold increased risk of NHL compared with uninfected controls [21]. This association was particularly strong for B-cell NHL subtypes, with diffuse large B-cell lymphoma and marginal zone lymphoma showing the most robust connections to HCV infection [22,23]. The consistency of this association across multiple studies and geographic regions lends credence to a potential causal relationship between HCV and NHL.

Beyond NHL, other hematologic malignancies have also been linked to HCV infection, albeit with varying degrees of evidence. A large-cohort study demonstrated an increased risk of multiple myeloma (odds ratio (OR)—1.98, 95% confidence interval (CI)—1.42–2.77) and acute myeloid leukemia (OR—1.54, 95% CI—1.05–2.26) in HCV-infected individuals [19]. These findings suggest that the impact of HCV on hematopoietic malignancies may extend beyond lymphoproliferative disorders.

The association between HCV and chronic lymphocytic leukemia remains a topic of debate in the scientific community. Some studies have reported an increased risk of chronic lymphocytic leukemia in HCV-infected patients [24,25], while others have found no significant association [26]. This discrepancy highlights the complexity of studying rare malignancies in the context of viral infections and underscores the need for large-scale, well-designed epidemiologic studies to elucidate these relationships.

### 3.2. Pancreatic Cancer

The potential link between HCV infection and pancreatic cancer has recently gained increasing attention. A nationwide study in Sweden found a 60% increased risk of pancreatic cancer among HCV-infected individuals [27]. This finding has been corroborated by other studies, including a meta-analysis, which reported a pooled relative risk of 1.26 (95% CI—1.03–1.50) for pancreatic cancer in HCV-infected patients [28]. The consistency of these results across different study designs and populations suggests that HCV infection may indeed be a risk factor for pancreatic cancer, although the underlying mechanisms remain to be fully elucidated.

### 3.3. Renal Cancer

Emerging evidence suggests a potential association between HCV infection and renal cell carcinoma. One study reported a 1.3-fold increased risk of kidney cancer among older HCV-infected individuals in the USA [19]. This finding was further supported by a systematic review and meta-analysis that found a pooled OR of 1.86 (95% CI: 1.11–3.11) for renal cell carcinoma in HCV-infected patients [29]. While these results are intriguing, additional research is needed to confirm this association and explore potential biological mechanisms.

### 3.4. Head and Neck Cancers

Several studies have reported an increased risk of head and neck cancers in HCV-infected individuals, particularly cancers of the oral cavity and oropharynx. There was a 1.4-fold increased risk of mid-pharyngeal cancer among HCV-infected patients [19]. A comprehensive meta-analysis reported a pooled odds ratio of 2.14 (95% CI 1.17–3.92) for oral cancer in HCV-infected patients [29]. These findings suggest that HCV infection may contribute to the development of head and neck malignancies, possibly through direct viral effects or indirect mechanisms related to chronic inflammation and immune dysregulation.

### 3.5. Intrahepatic Cholangiocarcinoma

While not strictly an extrahepatic cancer, intrahepatic cholangiocarcinoma has been consistently associated with HCV infection in multiple studies. The risk of this rare but aggressive malignancy appears to be 2–4-fold higher in HCV-infected patients compared with HCV-uninfected individuals [30,31]. A meta-analysis found a pooled relative risk of 3.42 (95% CI 1.96–5.99) for intrahepatic cholangiocarcinoma in HCV-infected individuals [32]. This strong association underscores the importance of considering biliary tract malignancies in the long-term management of HCV-infected patients.

### 3.6. Thyroid Cancer

The relationship between HCV infection and thyroid cancer remains less well-established compared with other malignancies. However, some studies have suggested a potential association. A large-cohort study found a hazard ratio of 1.68 (95% CI—1.18–2.40) for thyroid cancer in HCV-infected patients [33]. While this finding is intriguing, further research is needed to confirm the association and explore potential underlying mechanisms, such as HCV-induced thyroid autoimmunity or direct viral effects on thyroid tissue.

It is crucial to note that the strength of the association between HCV infection and extrahepatic cancers varies across studies and cancer types. Furthermore, the observed increased cancer risks in HCV-infected populations may be influenced by confounding factors, such as smoking, alcohol use, and other lifestyle factors [34,35]. Disentangling the direct effects of HCV infection from these potential confounders remains a significant challenge in epidemiologic research.

**Table 1 microorganisms-12-01926-t001:** Extrahepatic cancers associated with HCV infection.

Cancer Type	Risk Measure	Risk Increase (95% CI)	Key Findings	Study Type	Reference	Reference Number
Non-Hodgkin Lymphoma (NHL)	Relative Risk	2–3 fold increase	Strongest association with B-cell NHL subtypes	Meta-analysis	Dal Maso & Franceschi, 2006	[21]
Multiple Myeloma	Odds Ratio	1.98 (1.42–2.77)	Significant association in older HCV-infected individuals	Case-control study	Mahale et al., 2017	[19]
Acute Myeloid Leukemia	Odds Ratio	1.54 (1.05–2.26)	Increased risk in HCV-infected population	Case-control study	Mahale et al., 2017	[19]
Pancreatic Cancer	Relative Risk	1.26 (1.03–1.50)	Consistent association across multiple studies	Meta-analysis	Xu et al., 2013	[28]
Renal Cell Carcinoma	Odds Ratio	1.86 (1.11–3.11)	Potential association requiring further investigation	Meta-analysis	Ma et al., 2021	[29]
Oral Cancer	Odds Ratio	2.14 (1.17–3.92)	Significant association, particularly for oral cavity cancer	Meta-analysis	Ma et al., 2021	[29]
Thyroid Cancer	Hazard Ratio	1.68 (1.18–2.40)	Potential association requiring further confirmation	Cohort study	Antonelli et al., 2007	[33]

CI: confidence interval.

### 3.7. HCV Coinfection with Other Viral Hepatitis

The coinfection of HCV with other hepatitis viruses, particularly hepatitis B virus (HBV), can significantly increase the risk of both hepatic and extrahepatic cancers. HCV/HBV coinfection has been associated with a higher risk of hepatocellular carcinoma compared to infection with either virus alone [36]. Additionally, this coinfection may increase the risk of certain extrahepatic cancers, such as non-Hodgkin lymphoma, due to enhanced chronic immune stimulation and inflammation [37].

## 4. Mechanisms of HCV-Induced Extrahepatic Carcinogenesis (Figure 1)

The mechanisms by which hepatitis C virus (HCV) infection may contribute to extrahepatic cancer development are complex, multifaceted, and not fully elucidated. Researchers have proposed several potential pathways through which HCV might promote carcinogenesis outside the liver. These mechanisms often overlap and interact, creating a complex network of cellular and molecular alterations that can ultimately lead to the development of extrahepatic malignancies. The following section explores these proposed mechanisms in greater detail.

**Figure 1 microorganisms-12-01926-f001:**
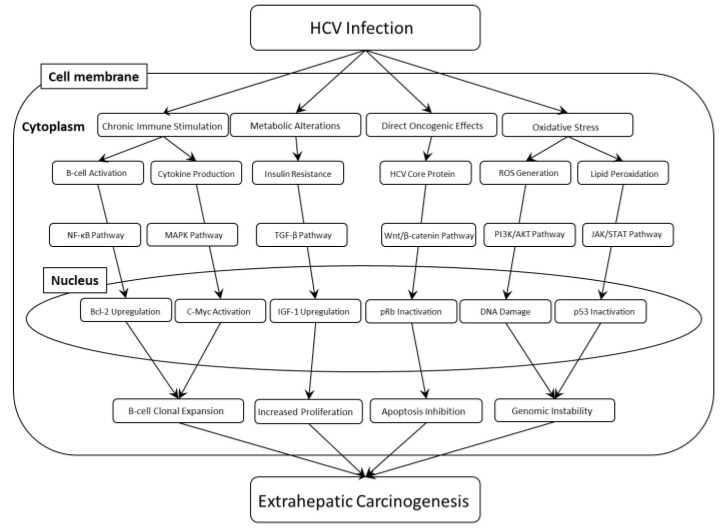
Mechanisms of HCV-induced extrahepatic carcinogenesis. HCV: hepatitis C virus. ROS: Reactive oxygen species. NF-κB: Nuclear Factor kappa-light-chain-enhancer of activated B cells. MAPK: mitogen-activated protein kinase. TGF-β: Transforming growth factor-β. PI3K/AKT: phosphatidylinositol-3 kinase/v-Akt Murine Thymoma Viral Oncogene. JAK/STAT: Janus kinase/signal transducers and activators of transcription. Bcl-2: B-cell lymphoma 2. IGF-1: Insulin-like Growth Factor 1. pRb: Retinoblastoma protein. DNA: deoxyribonucleic acid. p53: Tumor protein p53.

### 4.1. Chronic Immune Stimulation

Persistent HCV infection leads to chronic activation of the immune system, which may promote lymphoproliferation and the development of NHL [22,38]. The virus can directly infect and replicate in B-cells, potentially leading to clonal expansion and malignant transformation [39,40]. This continuous antigenic stimulation by HCV may result in the accumulation of genetic aberrations in B-cells over time, contributing to lymphomagenesis [41]. The chronic activation of B-cells can lead to the production of autoantibodies and cryoglobulins, further exacerbating the inflammatory response and potentially promoting the development of lymphoproliferative disorders.

### 4.2. Oxidative Stress

HCV infection induces significant oxidative stress, which can cause DNA damage and genomic instability in both hepatic and extrahepatic tissues [42,43]. The HCV core protein increases reactive oxygen species production and impairs mitochondrial function, contributing to oxidative stress-induced cellular damage [44]. This persistent state of oxidative stress can lead to the accumulation of genetic mutations, alterations in cellular signaling pathways, and the disruption of normal cellular processes. Over time, these changes may contribute to the initiation and progression of extrahepatic cancers in various organ systems.

### 4.3. Metabolic Alterations

HCV infection is associated with insulin resistance and altered lipid metabolism, which may contribute to the development of pancreatic cancer and other metabolic-related malignancies [45,46]. The virus interferes with insulin signaling pathways and promotes hepatic steatosis, leading to systemic metabolic disturbances that may increase cancer risk [47]. These metabolic alterations can affect multiple organ systems, potentially creating a pro-tumorigenic environment in extrahepatic tissues. For example, insulin resistance and hyperinsulinemia can promote cell proliferation and inhibit apoptosis, while dysregulated lipid metabolism may lead to the accumulation of lipotoxic intermediates that can damage cellular structures and contribute to carcinogenesis.

### 4.4. Direct Oncogenic Effects

Although less well-established compared with HCC, HCV proteins may have direct oncogenic effects on extrahepatic tissues [48]. For example, the HCV core protein interacts with cellular proteins involved in cell cycle regulation and apoptosis, such as p53 and p21 [49,50]. These interactions can disrupt normal cell cycle control mechanisms and promote uncontrolled cell proliferation. The NS3/4A protease has been implicated in interfering with DNA repair mechanisms, potentially contributing to genomic instability [51].

### 4.5. Inflammation and Fibrosis

Chronic inflammation and fibrosis induced by HCV infection may create a pro-tumorigenic microenvironment in various organs, potentially promoting cancer development [9,52]. The persistent inflammatory state leads to the production of pro-inflammatory cytokines and growth factors that can support tumor growth and progression [53]. This chronic inflammatory milieu can lead to the recruitment of immune cells, such as tumor-associated macrophages, which can further promote angiogenesis, tissue remodeling, and tumor growth. Moreover, the development of fibrosis in extrahepatic tissues can alter the local microenvironment, potentially creating niches that favor the survival and proliferation of transformed cells.

### 4.6. Immune Dysregulation

HCV infection can lead to alterations in immune function, including impaired natural killer (NK) cell activity and T-cell exhaustion, which may compromise tumor immunosurveillance [54,55]. HCV interferes with interferon signaling pathways, potentially impairing the host’s ability to mount an effective antitumor immune response [56]. This immune dysregulation can create a permissive environment for the development and progression of extrahepatic cancers. The impairment of NK cell function, in particular, may reduce the immune system’s ability to recognize and eliminate nascent tumor cells in extrahepatic tissues [56].

### 4.7. Extrahepatic Replication of HCV

HCV replication is not confined to the liver; accumulating evidence suggests it can occur in various extrahepatic tissues, potentially contributing to the development of extrahepatic cancers. Studies have demonstrated the presence of HCV RNA and proteins in peripheral blood mononuclear cells, including B and T lymphocytes, which may explain the increased risk of lymphoproliferative disorders observed in HCV-infected individuals [57]. Furthermore, HCV replication has been detected in the central nervous system, particularly in microglial cells and astrocytes, potentially linking to neurological complications and associated pathologies [58].

The virus has also been found to replicate in other organs, such as the salivary glands and thyroid gland. The presence of HCV RNA in salivary gland tissue may be associated with sialadenitis and an increased risk of salivary gland tumors [59]. Similarly, evidence of HCV replication in thyroid cells suggests a direct viral effect on thyroid pathology, potentially increasing the risk of thyroid dysfunction and cancer [60].

This extrahepatic replication underscores the systemic nature of HCV infection, and highlights the importance of comprehensive cancer screening and long-term follow-up in HCV-infected patients. Even after successful viral eradication with direct-acting antiviral therapy, vigilance for extrahepatic manifestations remains crucial due to the virus’ ability to affect multiple organ systems beyond the liver.

## 5. Impact of DAA Therapy on Extrahepatic Cancer Risk

The impact of different treatment approaches on extrahepatic cancer risk in HCV-infected patients has been a subject of extensive research. Table 2 summarizes the current understanding of extrahepatic cancer risk associated with different treatment strategies.

The risk profiles presented in Table 1 are based on several key studies. For HCV patients receiving no treatment, multiple studies have consistently shown an increased risk of extrahepatic cancers compared to the general population [19,20]. Traditional interferon-based treatments have been associated with a moderate decrease in extrahepatic cancer risk, particularly for lymphoproliferative disorders [61].

The impact of DAA treatment on extrahepatic cancer risk is still being elucidated. While initial studies raised concerns about a potential increase in cancer risk [11], more recent long-term data suggest that successful DAA treatment may actually decrease the risk of extrahepatic malignancies [62,63].

**Table 2 microorganisms-12-01926-t002:** Comparison of extrahepatic cancer risk.

Treatment Strategy	Overall Extrahepatic Cancer Risk	Specific Cancer Types	Relative Risk (95% CI)	Follow-Up Period	Key References	Reference Number
HCV—No Treatment	Increased	Non-Hodgkin Lymphoma Multiple Myeloma Head and Neck	2.24 (1.80–2.78) 1.97 (1.31–2.96) 1.56 (1.32–1.84)	Varied	Mahale et al. (2017)	[19]
		Non-Hodgkin Lymphoma	0.73 (0.55–0.96)			
HCV—Traditional (IFN/Ribavirin)	Moderately Decreased			Median 5.6 years	Kawamura et al. (2017)	[61]
		All Extrahepatic	0.81 (0.66–0.99)			
		All Extrahepatic	0.84 (0.73–0.96)			
HCV—DAA	Potentially Decreased	Hematological	0.64 (0.38–1.07)	Median 33.4 months	Carrat et al. (2019)	[62]
		Non-Hematological	0.86 (0.74–0.99)			

CI: confidence interval. HCV: hepatitis C virus. DAA: direct-acting antivirals.

### 5.1. Large Cohort Studies

In a nationwide cohort study in Sweden, 19,685 HCV-infected patients were included, of whom 4013 were treated with DAAs [64]. The study found no increased risk of extrahepatic cancer associated with DAA therapy after adjustment for age and comorbidity. However, there was an increased risk of extrahepatic cancer in DAA-treated patients compared with the general population without an HCV diagnosis, highlighting the need for further studies in aging populations with a history of HCV infection.

An analysis of US Census and National Center for Health Statistics data from 2007 to 2017 reported an increase in extrahepatic cancer mortality among HCV-infected individuals in the DAA era [65]. However, the study lacked individual-level therapy data, making it difficult to assess the direct association between DAAs and cancer risk.

Ioannou et al. (2019) examined the impact of HCV treatment on hematologic malignancies in a large cohort of US veterans [66]. The authors found that interferon-induced SVR was associated with a significant reduction in the risk of non-Hodgkin lymphoma, multiple myeloma, and overall hematologic malignancies. However, these associations were not observed with DAA-induced SVR during a mean follow-up of 2.9 years.

### 5.2. Registry-Based Studies

A registry-based study in the USA compared HCV-infected and untreated patients who received any type of HCV therapy, including interferon-based and DAA regimens, and found an 11% reduction in the incidence of extrahepatic cancer [67]. However, the association was not observed with DAA therapy alone, possibly owing to the short follow-up time for this group (12.7 months).

In a large prospective cohort study in France, SVR with DAA therapy was associated with low all-cause mortality, including both liver disease-related and non-liver disease-related deaths [62]. While the study did not specifically focus on extrahepatic cancer outcomes, it provides evidence for the overall health benefits of DAA therapy.

### 5.3. Meta-Analyses and Systematic Reviews

Masarone and Persico (2019) performed a systematic review and meta-analysis of studies examining the impact of HCV eradication on non-hepatocellular malignancies [68]. The authors found that achieving SVR was associated with a reduced risk of lymphoma and other hematologic malignancies. However, the analysis did not specifically focus on DAA-induced SVR owing to limited data.

A meta-analysis examined the impact of DAA treatment on extrahepatic manifestations of HCV infection, including cancer [69]. The study found a reduced risk of non-liver cancer (adjusted hazard ratio: 0.89) in patients who achieved vs. did not achieve SVR.

These studies highlight the complexity of assessing the impact of DAA therapy on extrahepatic cancer risk. The conflicting results may be attributed to several factors. (1) Differences in study design and populations: Studies vary in their design (e.g., retrospective vs. prospective), population characteristics, and geographic locations, which may contribute to discrepancies in the results [70]. (2) Length of follow-up: Many studies have relatively short follow-up periods, which may be insufficient to capture the long-term effects of DAA treatment on cancer risk [71]. (3) Confounding factors: HCV-infected individuals often have multiple risk factors for cancer, including smoking, alcohol use, and metabolic disorders. Adequately controlling for these confounders is challenging, and may lead to differences in results across studies [72]. (4) Competing risks: The dramatic reduction in liver-related mortality following DAA therapy may lead to an apparent increase in non-liver-related causes of death, including extrahepatic cancers, owing to longer survival times [73]. (5) Surveillance bias: Patients who receive vs. do not receive DAA therapy may undergo more frequent medical evaluations, potentially leading to the earlier detection of extrahepatic cancers [74].

### 5.4. Short-Term and Long-Term Effects of DAA Treatment on Extrahepatic Malignancy Risk

DAA treatment for HCV infection has shown complex effects on extrahepatic malignancy risk. While some studies initially reported a potential short-term increase in cancer diagnoses following DAA treatment, attributed to an “unmasking effect”, long-term data suggest an overall reduction in extrahepatic cancer risk [11]. However, long-term data suggest an overall reduction in extrahepatic cancer risk [5,62,63,75]. Figure 2 illustrates the conceptual trend of extrahepatic malignancy risk over time in DAA-treated patients compared to untreated HCV patients.

## 6. Immunological Considerations

The impact of DAA treatment on the immune system and its potential implications for cancer risk have been a subject of intense interest and ongoing research. The rapid viral clearance achieved by DAAs leads to significant changes in the immune landscape, which may have both beneficial and potentially concerning effects on cancer surveillance. Several studies have investigated the immunologic changes associated with DAA therapy, providing insights into the complex interplay between viral clearance, immune function, and cancer risk.

### 6.1. NK Cell Function

NK cells play a crucial role in tumor immunosurveillance and are often dysregulated in chronic HCV infection. It has been reported that successful DAA therapy significantly normalizes NK cell function [76,77]. This reduction in HCV-specific immune responses could potentially impact the immune system’s ability to recognize and eliminate HCV-associated malignant cells. The implications of this finding are twofold: while the finding may indicate a reduced inflammatory state, which is generally beneficial, it also raises concerns about the potential for decreased immunosurveillance against HCV-associated neoplastic cells. Further research is needed to determine whether this reduction in HCV-specific T-cell responses translates to an increased risk of HCC or other extrahepatic cancers long-term.

### 6.2. T-Cell Responses

T-cells, particularly clusters of differentiation 8+ cytotoxic T-cells, are critical components of the adaptive immune response against both viral infections and cancer. DAA-induced HCV clearance leads to a significant decrease in the strength of HCC-specific clusters of differentiation 8+ T-cell responses in cirrhotic patients [78]. This reduction in HCV-specific immune responses could potentially impact the immune system’s ability to recognize and eliminate HCV-associated malignant cells. The implications of this finding are twofold: while the finding may indicate a reduced inflammatory state, which is generally beneficial, it also raises concerns about the potential for decreased immunosurveillance against HCV-associated neoplastic cells. Further research is needed to determine whether this reduction in HCV-specific T-cell responses translates to an increased risk of HCC or other extrahepatic cancers long-term.

### 6.3. Cytokine Profiles

Several studies have investigated the impact of Direct-Acting Antiviral (DAA) treatment on cytokine profiles in HCV-infected patients. Hengst et al. performed a comprehensive analysis of serum cytokines and chemokines in patients who received DAA therapy [79]. The authors observed a rapid decline in interferon-stimulated genes and pro-inflammatory cytokines, including interferon-γ-induced protein-10, monocyte chemoattractant protein-1, and interleukin-18, within the first few days of therapy. This reduction in inflammatory mediators persisted throughout the therapeutic course and after achieving SVR. Burchill et al. (2017) reported similar findings, noting a significant decrease in plasma levels of interferon-γ-induced protein-10, macrophage inflammatory protein 1-beta, and tumor necrosis factor-alpha following DAA therapy [80]. Interestingly, the authors also observed a reduction in T-cell activation markers, suggesting a broader impact on the immune system beyond cytokine modulation.

DAA treatment leads to a rapid and sustained reduction in pro-inflammatory cytokines, potentially contributing to the resolution of systemic inflammation associated with chronic HCV infection. However, the long-term implications of these changes for immune surveillance and oncogenic risk require further investigation.

### 6.4. B Cell Abnormalities

Chronic HCV infection is associated with significant B-cell dysregulation, which may contribute to the increased risk of B-cell lymphoproliferative disorders observed in HCV-infected individuals. DAA treatment has been reported to markedly normalize the distribution and function of B-cell subsets in HCV-infected patients [81]. This restoration of B-cell homeostasis could potentially reduce the risk of B-cell lymphoproliferative disorders associated with chronic HCV infection. The normalization of B-cell function may have far-reaching implications for the immune system’s ability to mount effective humoral responses against emerging neoplastic cells, potentially contributing to improved cancer surveillance in the post-DAA therapy period. These immunologic changes highlight the complex and multifaceted interplay between viral clearance, immune function, and cancer risk. The rapid suppression of HCV replication by DAAs leads to a dramatically different immunological landscape compared with the gradual viral clearance achieved with interferon-based therapies [82]. This rapid shift in immune function may have both positive and negative implications for cancer risk. On the positive side, the reduction in chronic inflammation and restoration of normal immune cell function may reduce the risk of inflammation-driven carcinogenesis and improve overall immune surveillance. However, the rapid changes in immune function may also create a temporary state of immune dysregulation, potentially creating a window of vulnerability for emerging neoplastic cells.

## 7. Challenges in Studying Extrahepatic Cancer Risk in HCV-Infected Patients

Several challenges complicate the assessment of extrahepatic cancer risk in HCV-infected patients treated with DAAs. Many cancers have a long latency period, requiring extended follow-up to accurately assess risk. Most studies on DAA treatment have relatively short follow-up periods, which may be insufficient to capture long-term cancer outcomes [83,84]. The HCV-infected population treated with DAAs is often older and has more advanced liver disease compared with those treated in the interferon era, making direct comparisons between treatment eras challenging [63,85].

The dramatic reduction in liver-related mortality following DAA treatment may lead to an apparent increase in non-liver-related causes of death, including extrahepatic cancers, due to longer survival times [86,87]. HCV-infected individuals often have multiple risk factors for cancer, including smoking, alcohol use, and metabolic disorders. Adequately controlling for these confounders is challenging in observational studies [5,35,88].

For many extrahepatic cancers, there are limited data on their incidence in HCV-infected populations before the DAA era, making it difficult to establish baseline risks for comparison [89]. Additionally, the use of different study designs, populations, and outcome measures across studies makes it challenging to draw definitive conclusions about cancer risk [2,52]. These factors collectively contribute to the complexity of assessing the true impact of DAA treatment on extrahepatic cancer risk in HCV-infected patients.

## 8. Future Directions

To address the gaps in our understanding of extrahepatic cancer risk in HCV-infected patients treated with DAAs, several areas of research should be prioritized. Prospective cohort studies with extended follow-up periods (>5–10 years) are needed to assess the long-term impact of DAA treatment on extrahepatic cancer risk [86,90]. Leveraging national health registries and databases can provide the necessary sample sizes to detect changes in rare cancer outcomes [91].

More detailed studies focusing on specific cancer types, particularly those with established links to HCV infection (e.g., NHL, pancreatic cancer), are needed to elucidate the impact of DAA treatment on these malignancies [92,93].

Further research into the biological mechanisms underlying HCV-associated extrahepatic carcinogenesis and the impact of DAA treatment on these pathways is crucial [10,94,95,96].

Identifying subgroups of HCV-infected patients who may be at higher risk for extrahepatic cancers after DAA treatment could inform targeted surveillance strategies [96,97,98]. Risk factors for non-alcoholic fatty liver disease-associated HCC may be related to HCV-associated extrahepatic cancers, and common mechanisms of pathogenesis have been reported [99]. Evaluating the cost-effectiveness of cancer screening strategies in DAA-treated patients can help inform clinical guidelines and public health policies [100,101,102].

Further investigation into the long-term immunologic effects of DAA treatment and their potential impact on cancer risk is warranted [79,103,104]. This includes studying the restoration of immune function, changes in cytokine profiles, and the impact on tumor immunosurveillance mechanisms.

Identifying and validating biomarkers for the early detection of extrahepatic cancers in HCV-infected and DAA-treated patients could improve outcomes through earlier intervention [105,106,107]. This may involve exploring circulating tumor DNA, microRNAs, or other novel biomarkers specific to HCV-associated malignancies.

These research priorities will help us better understand the complex relationship between HCV infection, DAA therapy, and extrahepatic cancer risk, ultimately leading to improved patient care and outcomes.

## 9. Conclusions

While the current evidence does not suggest a notable increase in extrahepatic cancer risk associated with DAA treatment, ongoing surveillance and research are essential to fully elucidate the long-term oncological outcomes in this population. As our understanding of the complex interplay between HCV infection, immune function, and carcinogenesis continues to evolve, we may be able to develop more targeted and effective strategies for cancer prevention and early detection in individuals with a history of HCV infection [5,90].

## Figures and Tables

**Figure 2 microorganisms-12-01926-f002:**
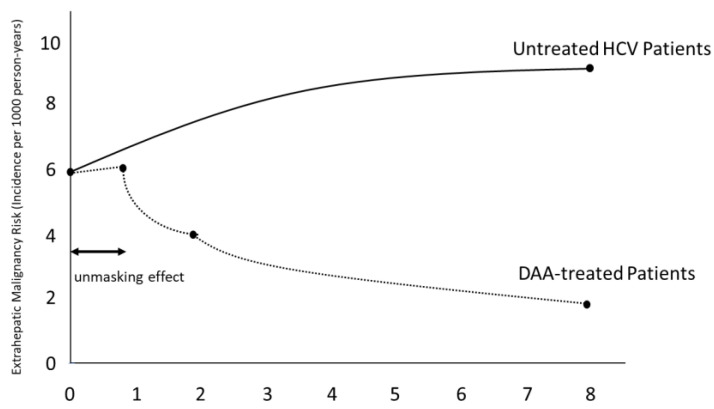
Impact of DAA treatment on extrahepatic malignancy risk. While short-term data may have shown an apparent increase in cancer diagnoses due to enhanced detection, long-term evidence supports a reduced risk of extrahepatic malignancies in successfully DAA-treated patients. HCV: hepatitis C virus. DAA: direct-acting antivirals.

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
