# Peer review of "Extrahepatic Cancer Risk in Patients with Hepatitis C Virus Infection Treated with Direct-Acting Antivirals"

_microorganisms, 2024, doi:10.3390/microorganisms12091926_

Round 1

Reviewer 1 Report

Comments and Suggestions for Authors

It is an interesting and comperhensive review discussing the risk of cancers beyond the liver in patients infected with chronic HCV infection taking DAA. The authors discussed possible extrahepatic cancers, mechanisms, and other factors such as immune cells, cytokine profile, etc.

Few points to be considered to improve the review

1) I would suggest the authors to include extraheptic replication of HCV as this represent the first evidence and risk factor of cancer development in these ares

2) Fig 1: I would suggest include siganlings pathwaying figures 

3) Authors should include a table that showed the difference between HCV-DAA, HCV-no treatmnet and HCV- traditional medication (IFN and ribavirin) in terms of extrahepatic cancer risk

4) What is the role of coinfection of HCV with other viral hepatitis in development of cancer?

Comments on the Quality of English Language

fine

Author Response

Prof. Dr. Nico Jehmlich

Editor-in-Chief

Microorganisms

17 Sep 2024

Dear Professor. Dr. Nico Jehmlich,

Manuscript ID: microorganisms-3210298

Type of manuscript: Review

Title: Extrahepatic Cancer Risk in Patients with Hepatitis C Virus Infection Treated with Direct-Acting Antivirals

Authors: Joji Tani *, Tsutomu Masaki, Oura Kyoko, Tomoko Tadokoro, Asahiro Morishita, and Hideki Kobara

Received: 31 Aug 2024

Thank you very much for your email of Sep 8, 2024, with the comments from the reviewer regarding our manuscript. We have revised our manuscript in accordance with these comments and have provided detailed responses below; revised text is shown in red. We hope that this satisfies the requirements for publication but are prepared to make additional changes if required.

                                                                                                                        Yours sincerely,

                                                                                                          Joji Tani, MD, PhD

Department of Gastroenterology and Neurology, Kagawa University Faculty of Medicine

1750-1 Ikenobe Miki-cho, Kita-gun, Kagawa 761-0793, Japan

Fax: +81-87-891-2158

Tel: +81-87-891-2156

Point-by-point discussion of the changes made in response to the review

【Comments1】

I would suggest the authors to include extraheptic replication of HCV as this represent the first evidence and risk factor of cancer development in these ares.

【Response1】

We have added Section 4.7 "Extrahepatic Replication of HCV", which discusses the evidence of HCV replication in lymphoid tissues, the central nervous system, salivary glands, and thyroid gland. This section highlights the potential link between extrahepatic replication and cancer development. I have added both the text and the citations directly into the manuscript, with each clearly highlighted in red.

4.7 Extrahepatic Replication of HCV

HCV replication is not confined to the liver; accumulating evidence suggests it can occur in various extrahepatic tissues, potentially contributing to the development of extrahepatic cancers. Studies have demonstrated the presence of HCV RNA and proteins in peripheral blood mononuclear cells, including B and T lymphocytes, which may explain the increased risk of lymphoproliferative disorders observed in HCV-infected individuals[57]. Furthermore, HCV replication has been detected in the central nervous system, particularly in microglial cells and astrocytes, potentially linking to neurological complications and associated pathologies[58].

The virus has also been found to replicate in other organs, such as the salivary glands and thyroid gland. The presence of HCV RNA in salivary gland tissue may be associated with sialadenitis and an increased risk of salivary gland tumors[59]. Similarly, evidence of HCV replication in thyroid cells suggests a direct viral effect on thyroid pathology, potentially increasing the risk of thyroid dysfunction and cancer[60].

This extrahepatic replication underscores the systemic nature of HCV infection and highlights the importance of comprehensive cancer screening and long-term follow-up in HCV-infected patients. Even after successful viral eradication with direct-acting antiviral therapy, vigilance for extrahepatic manifestations remains crucial due to the virus's ability to affect multiple organ systems beyond the liver.

  1. Revie, D.; Salahuddin, S.Z. Human cell types important for hepatitis C virus replication in vivo and in vitro: old assertions and current evidence. Virol J 2011, 8, 346, doi:10.1186/1743-422x-8-346.
  2. Fletcher, N.F.; McKeating, J.A. Hepatitis C virus and the brain. J Viral Hepat 2012, 19, 301-306, doi:10.1111/j.1365-2893.2012.01591.x.
  3. Gonzalez-Moles, M.A.; Bravo, M.; Gonzalez-Ruiz, L.; Ramos, P.; Gil-Montoya, J.A. Outcomes of oral lichen planus and oral lichenoid lesions treated with topical corticosteroid. Oral Dis 2018, 24, 573-579, doi:10.1111/odi.12803.
  4. Mistry, P.K. Rare disease clinical research network's urea cycle consortium delivers a successful clinical trial to improve alternate pathway therapy. Hepatology 2013, 57, 2100-2102, doi:10.1002/hep.26106.

【Comments2】

Fig 1: I would suggest include siganlings pathwaying figures

【Response2】

The updated Figure 1 now includes key signaling pathways such as NF-κB, MAPK, and PI3K/AKT, illustrating how HCV infection affects these cellular processes and potentially contributes to carcinogenesis.

【Comments3】

  1. Authors should include a table that showed the difference between HCV-DAA, HCV-no treatmnet and HCV- traditional medication (IFN and ribavirin) in terms of extrahepatic cancer risk.

【Response3】

We have changed Section “5. Impact of DAA Treatment on Extrahepatic Cancer Risk”. The impact of different treatment approaches on extrahepatic cancer risk in HCV-infected patients has been a subject of extensive research. Table 2 summarizes the current understanding of extrahepatic cancer risk associated with different treatment strategies.

We have included a new Table 2 comparing the extrahepatic cancer risk among HCV-DAA, HCV-no treatment, and HCV-traditional medication (IFN and ribavirin) groups.

The impact of different treatment approaches on extrahepatic cancer risk in HCV-infected patients has been a subject of extensive research. Table 2 summarizes the current understanding of extrahepatic cancer risk associated with different treatment strategies.

The risk profiles presented in Table 1 are based on several key studies. For HCV patients receiving no treatment, multiple studies have consistently shown an increased risk of extrahepatic cancers compared to the general population[19,20]. Traditional interferon-based treatments have been associated with a moderate decrease in extrahepatic cancer risk, particularly for lymphoproliferative disorders[61].

The impact of DAA treatment on extrahepatic cancer risk is still being elucidated. While initial studies raised concerns about a potential increase in cancer risk[11], more recent long-term data suggest that successful DAA treatment may actually decrease the risk of extrahepatic malignancies[62,63].

  1. Kawamura, Y.; Ikeda, K.; Arase, Y.; Yatsuji, H.; Sezaki, H.; Hosaka, T.; Akuta, N.; Kobayashi, M.; Suzuki, F.; Suzuki, Y.; et al. Viral elimination reduces incidence of malignant lymphoma in patients with hepatitis C. Am J Med 2007, 120, 1034-1041, doi:10.1016/j.amjmed.2007.06.022.

【Comments1】

  1. What is the role of coinfection of HCV with other viral hepatitis in development of cancer?

【Response4】

In the new subsection 3.7, we discuss how HCV coinfection with other hepatitis viruses, particularly HBV, can increase the risk of both hepatic and extrahepatic cancers. We cite relevant studies demonstrating the synergistic effect of coinfection on cancer development. I have added both the text and the citations directly into the manuscript, with each clearly highlighted in red.

3.7 HCV Coinfection with Other Viral Hepatitis

Coinfection of HCV with other hepatitis viruses, particularly hepatitis B virus (HBV), can significantly increase the risk of both hepatic and extrahepatic cancers. HCV/HBV coinfection has been associated with a higher risk of hepatocellular carcinoma compared to infection with either virus alone[36]. Additionally, this coinfection may increase the risk of certain extrahepatic cancers, such as non-Hodgkin lymphoma, due to enhanced chronic immune stimulation and inflammation[37].

  1. Cho, L.Y.; Yang, J.J.; Ko, K.P.; Park, B.; Shin, A.; Lim, M.K.; Oh, J.K.; Park, S.; Kim, Y.J.; Shin, H.R.; et al. Coinfection of hepatitis B and C viruses and risk of hepatocellular carcinoma: systematic review and meta-analysis. Int J Cancer 2011, 128, 176-184, doi:10.1002/ijc.25321.
  2. Marcucci, F.; Mele, A. Hepatitis viruses and non-Hodgkin lymphoma: epidemiology, mechanisms of tumorigenesis, and therapeutic opportunities. Blood 2011, 117, 1792-1798, doi:10.1182/blood-2010-06-275818.

Reviewer 2 Report

Comments and Suggestions for Authors

After reading the article I still be confused whether the DAA treatment decreased or increased the extra hepatic malignancy. Maybe you could draw a picture to describe it.

Comments on the Quality of English Language

Nil

Author Response

Prof. Dr. Nico Jehmlich

Editor-in-Chief

Microorganisms

17 Sep 2024

Dear Professor. Dr. Nico Jehmlich,

Manuscript ID: microorganisms-3210298

Type of manuscript: Review

Title: Extrahepatic Cancer Risk in Patients with Hepatitis C Virus Infection Treated with Direct-Acting Antivirals

Authors: Joji Tani *, Tsutomu Masaki, Oura Kyoko, Tomoko Tadokoro, Asahiro Morishita, and Hideki Kobara

Received: 31 Aug 2024

Thank you very much for your email of Sep 8, 2024, with the comments from the reviewer regarding our manuscript. We have revised our manuscript in accordance with these comments and have provided detailed responses below; revised text is shown in red. We hope that this satisfies the requirements for publication but are prepared to make additional changes if required.

                                                                                                                        Yours sincerely,

                                                                                                          Joji Tani, MD, PhD

Department of Gastroenterology and Neurology, Kagawa University Faculty of Medicine

1750-1 Ikenobe Miki-cho, Kita-gun, Kagawa 761-0793, Japan

Fax: +81-87-891-2158

Tel: +81-87-891-2156

Point-by-point discussion of the changes made in response to the review

【Comments1】

After reading the article I still be confused whether the DAA treatment decreased or increased the extra hepatic malignancy. Maybe you could draw a picture to describe it.

【Response1】

We appreciate your feedback regarding the clarity of our discussion on the effects of DAA treatment on extrahepatic malignancy risk. To address your concerns, we have added a new section 5.4 titled "Short-term and Long-term Effects of DAA Treatment on Extrahepatic Malignancy Risk". This section provides a concise summary of the current evidence on this topic. Additionally, to further clarify the impact of DAA treatment on extrahepatic malignancy risk, we have added a new Figure 2 that visually illustrates the long-term trends in extrahepatic cancer risk following DAA treatment.

5.4 Short-term and Long-term Effects of DAA Treatment on Extrahepatic Malignancy Risk

 DAA treatment for HCV infection has shown complex effects on extrahepatic malignancy risk. While some studies initially reported a potential short-term increase in cancer diagnoses following DAA treatment, attributed to an "unmasking effect," long-term data suggest an overall reduction in extrahepatic cancer risk[11]. However, long-term data suggest an overall reduction in extrahepatic cancer risk[5,62,63,75]. Figure 2 illustrates the conceptual trend of extrahepatic malignancy risk over time in DAA-treated patients compared to untreated HCV patients.

Reviewer 3 Report

Comments and Suggestions for Authors

The authors systematically reviewed numerous articles to provide a broad view on the relationship between Hepatitis C virus (HCV) infection and extrahepatic cancer for the patients treated with direct-acting antivirals (DAAs). This review covers many aspects related to this topic, and it is especially considerate to also include researches on HCV-associated immune modifications. Overall, I think it is a quality work that is worth publishing. I only have several minor suggestions:

Section 3.2 is repeated.

I personally think that section 9 is redundant and could be deleted.

It would be much appreciated if Figure 1, the only figure in this review, could be more visually pleasing (a schematic at the cellular level, for example).

Author Response

Prof. Dr. Nico Jehmlich

Editor-in-Chief

Microorganisms

17 Sep 2024

Dear Professor. Dr. Nico Jehmlich,

Manuscript ID: microorganisms-3210298

Type of manuscript: Review

Title: Extrahepatic Cancer Risk in Patients with Hepatitis C Virus Infection Treated with Direct-Acting Antivirals

Authors: Joji Tani *, Tsutomu Masaki, Oura Kyoko, Tomoko Tadokoro, Asahiro Morishita, and Hideki Kobara

Received: 31 Aug 2024

Thank you very much for your email of Sep 8, 2024, with the comments from the reviewer regarding our manuscript. We have revised our manuscript in accordance with these comments and have provided detailed responses below; revised text is shown in red. We hope that this satisfies the requirements for publication but are prepared to make additional changes if required.

                                                                                                                        Yours sincerely,

                                                                                                          Joji Tani, MD, PhD

Department of Gastroenterology and Neurology, Kagawa University Faculty of Medicine

1750-1 Ikenobe Miki-cho, Kita-gun, Kagawa 761-0793, Japan

Fax: +81-87-891-2158

Tel: +81-87-891-2156

Point-by-point discussion of the changes made in response to the review

【Comments1】

Section 3.2 is repeated. I personally think that section 9 is redundant and could be deleted.

【Response1】

We have streamlined the manuscript by removing the repeated Section 3.2 and integrating relevant information from the deleted Section 9 into other parts of the paper.

3.2 Pancreatic Cancer

The potential link between HCV infection and pancreatic cancer has recently gained increasing attention. A nationwide study in Sweden found a 60% increased risk of pancreatic cancer among HCV-infected individuals[27]. This finding has been corroborated by other studies, including a meta-analysis, which reported a pooled relative risk of 1.26 (95% CI: 1.03–1.50) for pancreatic cancer in HCV-infected patients[28].

【Comments2】

It would be much appreciated if Figure 1, the only figure in this review, could be more visually pleasing (a schematic at the cellular level, for example).

【Response2】

The updated Figure 1 now includes key signaling pathways such as NF-κB, MAPK, and PI3K/AKT, illustrating how HCV infection affects these cellular processes and potentially contributes to carcinogenesis.

Round 2

Reviewer 1 Report

Comments and Suggestions for Authors

The authors addressed my comments

Comments on the Quality of English Language

Fine

Reviewer 2 Report

Comments and Suggestions for Authors

Nil

Comments on the Quality of English Language

Nil